# Atypical Pupil Reactions in Brain Dead Patients

**DOI:** 10.3390/brainsci11091194

**Published:** 2021-09-10

**Authors:** Joanna Sołek-Pastuszka, Małgorzata Zegan-Barańska, Jowita Biernawska, Marcin Sawicki, Waldemar Iwańczuk, Kornel Chełstowski, Romuald Bohatyrewicz, Wojciech Dąbrowski, Klaudyna Kojder

**Affiliations:** 1Clinic of Anaesthesiology and Intensive Care, Pomeranian Medical University in Szczecin, 71-252 Szczecin, Poland; pastuszka@mp.pl (J.S.-P.); lisienko@wp.pl (J.B.); romuald.bohatyrewicz@pum.edu.pl (R.B.); klaudynakojder@gmail.com (K.K.); 2Department of Anesthesiology, Intensive Therapy and Acute Intoxications, Pomeranian Medical University in Szczecin, 70-111 Szczecin, Poland; 3Department of Diagnostic Imaging and Interventional Radiology, Pomeranian Medical University in Szczecin, 71-252 Szczecin, Poland; msaw@pum.edu.pl; 4Department of Anaesthesiology and Intensive Therapy, Regional Hospital in Kalisz, 62-800 Kalisz, Poland; iwanczuk.waldemar@gazeta.pl; 5Department of Laboratory Diagnostics, Pomeranian Medical University in Szczecin, 70-111 Szczecin, Poland; kornelch@pum.edu.pl; 6Department of Anaesthesiology and Intensive Therapy, Medical University, 20-090 Lublin, Poland; w.dabrowski5@gmail.com

**Keywords:** pupillometer, brain death, pupillary reactivity, pupillary size, ciliospinal reflex

## Abstract

Background: During routine diagnosis of brain death, changes in pupil diameter in response to the stimulation of peripheral nerves are sometimes observed. For example, pupillary dilation after diagnosed brain death is described in the literature as the ciliospinal reflex. However, pupil constriction creates diagnostic doubts. Objective: The pupillometric analysis of pupil response to stimulation of the cervicothoracic spinal cord in patients with diagnosed brain death. Methods: Instrumental tests to confirm the arrest of cerebral circulation were performed in 30 adult subjects (mean age 53.5 years, range 26–75 years) with diagnosed brain death. In addition, a pupillometer was used to measure the change in pupil diameter in response to neck flexion. Intervention: Flexion of the neck and measuring the response in change of the pupil with the use of the pupillometer. Results: The change in the pupil was observed in the examined group of patients. Difference in pupil size ≥ 0.2 mm was observed in 14 cases (46%). In five cases (17%), pupil constriction was found (from 0.2 to 0.7 mm). Measurement error was +/− 0.1 mm. Conclusions: Both pupillary constriction and dilatation may occur due to a ciliospinal reflex in patients with brain death. This phenomenon needs further research in order to establish its pathophysiology.

## 1. Introduction

The diagnosis of brain death (BD) is an equivalent to irreversible cessation of all brain function. Over the past few decades, an improvement in the determination ofBD was made. The evaluation of BD is based on rigorous clinical examination. According to the protocol, the evaluation consists of two series of clinical examinations; moreover, in problematic cases, such as craniofacial injuries, infratentorial pathology or unusual reflexes, ancillary tests are performed. Determination of the brainstem reflexes is based on examination of the pupils’ reaction to bright light, the corneal reflexes, oculovestibular reflexes, cough and gag reflexes, oculocephalic reflexes, and facial movement to noxious stimuli at supraorbital nerve and the apnea test [1]. An assessment of brain reflexes is performed in each series of clinical examination, separated by a specified time period. It is well known that during an evaluation, various spinal reflexes can occur.

During the series of clinical examinations of BD patients, pupillary dilatation after neck flexion was observed by the authors. The diagnosis of BD was confirmed in this case not only by the mandatory series of clinical investigation and apnea tests, but also by the ancillary test because of the unusual reflexes. No data were found in the literature to describe this unusual behavior of pupils during BD evaluation.

A literature analysis suggests a possible explanation: after the stimulation of the nerves originating from the cervicothoracic part of the spinal cord, the spinal part of the sympathetic system could be activated and thus, be responsible for pupillary dilation in ciliospinal reflex, as shown in Figure 1 [2]. During the oculocephalic reflex test, a small change in pupil diameter may be unnoticeable or difficult to assess. It is important to note that minimal movements of already dilated pupils can be overlooked during an examination. A pupillometer should be used to increase accuracy and obtain more precise results, as well as detect even the smallest (0.1 mm) changes in pupil diameter [3,4]. We expect that the aforementioned reaction might cause hesitation during BD certification, and as a result, potentially delay BD diagnosis [5]. The aim of this study is to exam the reaction of the pupils after a neck flexion in patients with declared BD.

## 2. Materials and Methods

Thirty adult subjects with declared BD were included in the study. Brain death certification was performed according to the national protocol [6]. Patients with facial or orbital injury were excluded. Brain death in all cases was confirmed with ancillary tests: transcranial Doppler (TCD), electroencephalography (EEG) or digital subtraction angiography (DSA). Pupil measurements were taken immediately after the diagnosis of BD. The measurements of pupil diameter were taken twice: A—before neck flexion; B—after neck flexion. We used one type of tool, namely, the AlgiScan (Equip Medkey BV) pupillometer. It is a handheld instrument with a built-in digital camera that records very small changes in diameter. Considering that the pupilometer measurement error is +/− 0.1 mm, it was assumed that a difference in pupil size of ≥0.2 mm would be significant. The study protocol received decision no. (KB-0012/116/13) from the Bioethical Committee of the Pomeranian Medical University in Szczecin, Poland, and was also registered at clinicaltrials.gov (6 September 2021) with the identifier: NCT03281993.

## 3. Results

Eighteen male and twelve female patients were included in the study. The age range was 26–75 years, with the mean age 56. Presentation time of areflexia was different among the patients, from 6 to 96 hours. The reason for BD included cerebrovascular accident (CVA) in 25 cases (83%) and traumatic brain injury (TBI) in 5 cases (17%). In 14 cases, a change in pupil size of ≥0.2 mm was observed, of which, in 5 cases, constriction ≥ 0.2 mm was noted, and in 8 cases, dilatation ≥ 0.2 mm was detected. Only one case presented constriction in the left and dilatation in the right eye. In the rest, the change in pupil size was <0.2 mm. All data are presented in Table 1.

## 4. Discussion

This is the first published case series investigating the unusual pupil movements after neck flexion in BD-determined patients. The examined series of patients revealed discrete variation in pupil size after applying neck flexion.

Damage to several levels of the pupil control centers occurs as a result for BD. The diameter of a denervated pupil, without any parasympathetic and sympathetic control, remains approximately 3–7 mm dilated, according to various works in the literature [7,8]. At the onset of BD, together with nuclei necrosis of the oculomotor nerve, parasympathetic impulsion always wanes, leaving the pupil dilated at least 3–4 mm. However, spontaneous sympathetic impulsation from the ciliospinal center may be partially preserved. The reason for this phenomenon is sympathetic impulsion, associated with the restitution of the spinal cord systems, which occurs after BD, yet after some time, and usually together with the termination of spinal cord shock. It is manifested by the restoration of neurogenic vessels’ tension and their ability to induce sympathetic reflexes. This impulsation is conducted throughout a reflex arch, which begins in flexed neck, is passed through spinal cord and ends in pupils, omitting ways throughout the brainstem Figure 1.

Takeuschi et al., in a study on 628 patients with BD, found pupil diameters below 4 mm in only in 4.3% of them [9]. Olgun et al. examined pupil diameter in 11 children (mean age 3.9 years) and 17 adults (mean age 52 years) with determined BD [7]. The authors pointed out that pupils are significantly dilated in children in comparison to adults. A similar analysis was presented by Sagishima et al. [8].

In addition, partial vivacity of isolated peripheral structures of the central nervous system in BD is always possible due to the preserved minimal blood flow to the intracranial space, confirmed by computed tomographic angiography, which is well described in the literature [10].

Most of the authors suggest a pupil diameter of ≥ 4 mm as a diagnostically accepted criterion for BD, with no reaction to light observed at the same time. The application of this assumption has been reflection in the protocol of BD certification (e.g., Australia-New Zealand, USA, Japan) [11,12,13]. This proposition has disadvantages, because it may lead to reduced sensitivity of diagnostic protocols. Reshi et al. examined pupil diameter in 41 patients, approximately 14 hours from the determination of BD [14]. Pupil diameter fluctuated between 2.26 mm and 6.71 mm. The authors suggest performing an ancillary test if the diameter of the pupil is less than 4 mm. Pupil size below 4 mm is the cause of reduced sensitivity of the diagnostic protocol below 30% [14]. Most guidelines do not mention the ciliospinal reflex as a one of the BD criteria. The exceptions are Japanese, Austrian and Swiss guidelines, in which the existence of this reflex excludes BD [12,15]. Prof. E. Schmutzhard, one of the editors of the Austrian recommendations, explained inclusion of this restriction into guidelines based on examples in which pupils are dilated in a response to pain stimuli within the scope of innervations of trigeminal nerves tension. However, the existence of the ciliospinal reflex as a reaction to core nerves stimulation by receptors does not exclude BD. To the best of our knowledge, there are no references to a pupillometer in such cases in the existing literature. Usually in clinical practice, pupillary assessment is based upon visual examination using a manual source of lights and the evaluation remains subjective. This routine examination has fundamental diagnostic and prognostic values. Therefore, use of the pupilometer technique allows clinicians to have objective and accurate assessments of pupil size and reactivity to light and limit biases [16,17].

As long as the reaction of pupil dilation in response to head flexion after BD is fully explainable by the existence of the ciliospinal reflex, the interpretation of the reaction of miosis is particularly difficult. Since, in these cases, BD was confirmed by ancillary tests, one should first exclude the involvement of brainstem centers and accept the peripheral type of this phenomenon. Shlugman et al. presented a case of alternate and spontaneous constriction and dilation of pupils [18]. The time of gradual constriction in one pupil lasted about 3 seconds, while the second pupil was dilated. This situation was repeated within approximately 5-minute intervals, in which pupils were ‘mid-dilated’ and indifferent to light in direct and consensual reaction. In this case, BD was confirmed in electroencephalographic examination. The authors suggest that the phenomenon mentioned above was probably the result of the reaction of the denervated, hypersensitive pupil constrictor muscle to circulating biogenic amine and neurotransmitters. As already mentioned, after BD, the centripetal path of the pupil dilation reflex is damaged at the level of the brain stem in response to light. However, the centrifugal path is partially preserved. The interchanging pupils’ reaction is difficult to explain since it indicates the existence of reverse feedback between innervations of the pupil muscles. The only integration level which may exist in this situation is the feedback at the level of the ciliospinal center.

## 5. Conclusions

Both pupillary constriction and dilatation may occur due to a ciliospinal reflex in patients with brain death. This phenomenon needs further research in order to establish its pathophysiology. A pupilometer is a valuable tool to reveal minimal pupil movements and is more precise than standard measurements, thereby avoiding the intrinsic limitations of manual subjective examination.

## Figures and Tables

**Figure 1 brainsci-11-01194-f001:**
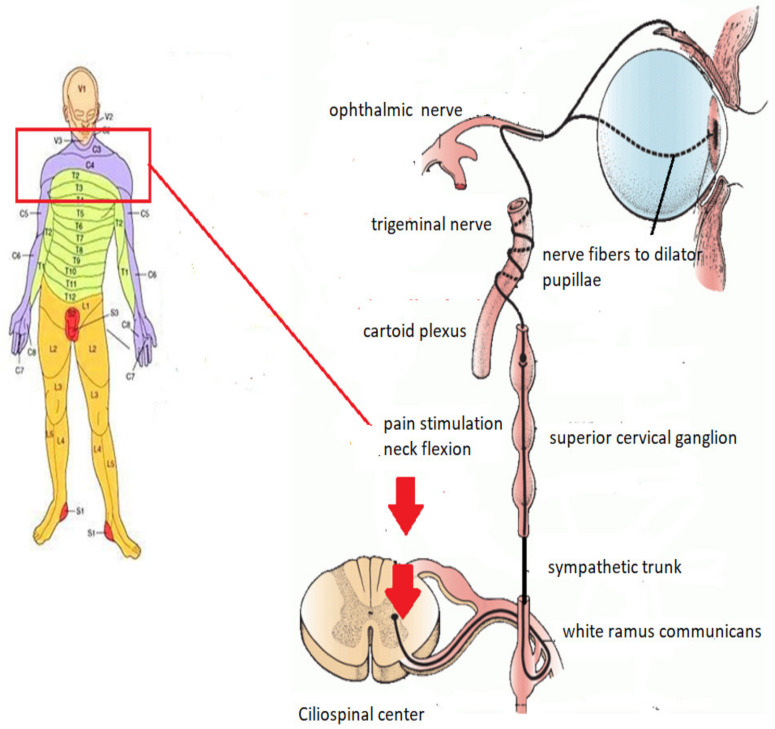
Ciliospinal reflex pathway.

**Table 1 brainsci-11-01194-t001:** Diameter of pupils (mm): A—before head flexion, B—after head flexion, TBI—trauma brain injury, CVA—cerebrovascular, TCD—transcranial Doppler, EEG—electroencephalography, DSA—digital subtraction angiography, F—female, M—male. All pupil size measurements are provided in millimeters (mm).

No.	Gender	Age	Cause of Brain Death	Time after Onset of Areflexia	Ancillary Test	Left Pupil A (mm)	Left Pupil B (mm)	Left Pupil Difference (mm)	Right Pupil A (mm)	Right Pupil B (mm)	Right Pupil Difference (mm)
1.	F	61	CVA	43	TCD	5.8	6.4	0.6	5.9	6.5	0.6
2.	M	57	CVA	20	DSA	5.2	5.4	0.2	5.7	5.8	0.1
3.	M	59	CVA	12	TCD	4.5	4.7	0.2	4.4	4.5	0.1
4.	F	48	CVA	6	TCD	6.8	6.5	−0.3	6.6	10	3.4
5.	M	61	CVA	13	TCD	3.2	3.4	0.2	2.9	2.9	0
6.	F	49	CVA	10	TCD	5.1	5.2	0.1	5.6	5.8	0.2
7.	M	53	CVA	37	EEG	3.6	4.4	0.8	3.4	4.3	0.9
8.	F	44	CVA	21	TCD	6.5	6.3	−0.2	6.2	6.2	0
9.	M	73	TBI	24	TCD	5.8	5.2	−0.6	5.1	5	−0.1
10.	M	33	CVA	56	TCD	4.9	4.9	0	4.9	5.1	0.2
11.	M	26	CVA	26	TCD	6	5.3	−0.7	5.5	5.6	0.1
12.	M	66	CVA	21	TCD	3.9	4.1	0.2	3.9	4.1	0.2
13.	F	75	CVA	23	TCD	5.3	5.1	−0.2	5	5	0
14.	M	44	CVA	28	TCD	6.2	6.2	0	5.9	6.1	0.2

## Data Availability

Clinic of Anaesthesiology and Intensive Care, Pomeranian Medical University in Szczecin, 71-252 Szczecin, Poland.

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
