# Peer review of "Atypical Pupil Reactions in Brain Dead Patients"

_brainsci, 2021, doi:10.3390/brainsci11091194_

Round 1

Reviewer 1 Report

Dear Authors, thank you for giving me the opportunity to read this work. I think it is interesting. However there are some aspects that you should address.

Title: I think you should add "Case series" in the title.

Introduction: Page 2 line 47: please, reformulate the sentence, it is too questionable.

Page 2 line 53:"...literature analysis..." please, add reference.

As a general rule, at least 5 references should be included in the introduction.

Methods: do you have an Ethic Committee number? Please, add it.

You did not calculate a sample size: why? Because this is a case series study?

Results: table 1--> please, add the unit of measures

I think that you could improve this paper by adding a figure showing the supposed ciliospinal reflex path...

Discussion: please widen the importance of pupillometer for detection of pupil size and response to light stimulus.

I'm looking to read a revised version of the manuscript.

Best regards

Author Response

Reviewer 1

Dear Reviewer, thank you for valuable comments to our manuscript. According to your suggestion all amendments were made.

Title: I think you should add "Case series" in the title. - Following your advice,  we changed it.

Introduction: Page 2 line 47: please, reformulate the sentence, it is too questionable.- The sentence was reformulate.

Page 2 line 53:"...literature analysis..." please, add reference. – Reference was added (line 55)

As a general rule, at least 5 references should be included in the introduction. – After added above ref altogether is five.

Methods: do you have an Ethic Committee number? Please, add it.- The EC no was added.

You did not calculate a sample size: why? Because this is a case series study?-Changed as suggested in first sentence.

Results: table 1--> please, add the unit of measures – The unit was added.

I think that you could improve this paper by adding a figure showing the supposed ciliospinal reflex path...-Was added

Discussion: please widen the importance of pupillometer for detection of pupil size and response to light stimulus. – Additional information was added  (line 134-139 with references and 162-164).

Kind regards,

Authors.

Reviewer 2 Report

Dear all,

the authors of the manuscript "Atypical pupil reactions in patients with brain death" evaluate the pupil reactions of 30 adults with verified brain death. They document that minimal pupil reactions are possible (despite a certified brain death) and provide an hypothesis on the mechanism.

The manuscript is well written and plausible. However, the results are sparse. The study type should be changed (article instead of protocol). Further, in my opinion the significance of the content (in terms of clinical relevance) is low.

Author Response

Dear Reviewer,

Thank you for your comments to our manuscript. \we change the type of the manuscript to a case series.

It is difficult for us to agree with given opinion regarding significance of our manuscript. In this manuscript we would like to not only underline the importance of emerging one of the phenomena during  examination of BD patient, but also by providing ours observations could help to recognise this problematic issue in BD patients, as well as offer a basis for further research and emphasize the usefulness of  pupilometers as a tool for precise measurement.

With kind regards,

Authors.

Round 2

Reviewer 1 Report

None